# A one-step foraminoplasty via a large trephine in percutaneous endoscopic transforaminal discectomy for the treatment of lumbar disc herniation

**Zhaoyu Yu** [1,2], **Yao Lu**[2], **Yong Li**[2], **Yan An**[3]*, **Bo Wang** [3]*

**1** The Second Clinical Medicine College, Guangzhou University of Chinese Medicine, Guangzhou, P.R. China, **2** Guangdong Province Hospital of Traditional Chinese Medicine Zhuhai Branch, Zhuhai, P.R. China, **3** Department of Spine Surgery, Beijing Jishuitan Hospital, Beijing, P.R. China

\* drwangbo@pku.edu.cn (BW); ay_jstspine@vip.163.com (YA)

## Abstract

### Background

Transforaminal percutaneous endoscopic lumbar discectomy (PELD) is a widely used basic technique for lumbar disc herniation (LDH) with advantages including causing less trauma and fast recovery. The secure, efficient, and rapid enlargement of the intervertebral foramen is a key step in PELD procedures. However, the conventional multi-step trephine system for foraminoplasty involves complicated surgical procedures. In this study, we reported an improved one-step foraminoplasty via a large trephine with simplified surgical procedures, reduced radiation exposure, and shortened operative time.

### Methods

70 LDH patients who underwent PELD were retrospectively reviewed in this study. The conventional multi-step trephine system was used for foraminoplasty in 35 patients in the multi-step (MS) group, and the single large trephine was used in the other 35 patients in the one-step (OS) group. Indicators including the operative time, the time to establish the working cannula, intraoperative fluoroscopy times, the radiation dose, and postoperative complications were compared between the MS and OS group.

### Results

The operative time and the time to establish the working cannula in the OS group was significantly shorter than that in the MS group (P < 0.01); intraoperative fluoroscopy times and the radiation dose in the OS group were significantly smaller than those in the MS group (P < 0.01). There was no statistical difference in the incidence of postoperative complications between the two groups (P > 0.05). The postoperative VAS scores and ODI scores (2 days and 3 months after the surgery) were significantly lower than the preoperative scores in both groups (P < 0.01), and there was no statistical difference in VAS scores or ODI scores between the two groups at the same time points (P > 0.05).

**Data Availability Statement:** All relevant data are within the manuscript and its Supporting information files.

**Funding:** The author(s) received no specific funding for this work.

**Competing interests:** The authors have declared that no competing interests exist.

## Conclusions

The one-step foraminoplasty via a single large trephine is an optimized technique evolving from the conventional multi-step foraminoplasty, showing significant superiority in simplified operation, shorted operative time, and reduced radiation exposure.

## 1. Introduction

Endoscopic spinal surgery has become the mainstream minimally invasive spine surgery (MISS) technique for lumbar disc herniation (LDH) [1]. Transforaminal percutaneous endoscopic lumbar discectomy (PELD) is a widely used basic technique with advantages including causing less trauma, improved surgical outcomes, fast recovery, and restored spinal stability. PELD requires the successful placement of a working cannula through the intervertebral foramen, so the secure, efficient, and rapid enlargement of the intervertebral foramen is a key step in surgical procedures [2]. The conventional multi-step trephine system is a frequently used tool for foraminal enlargement [3], but it involves complicated surgical procedures, a long learning curve, and multiple fluoroscopies that expose patients and surgeons to high radiation [4, 5]. Therefore, we improved the conventional procedures of foraminoplasty via the use of a single large trephine, which simplified the complicated surgical procedures and reduced the radiation exposure while achieving satisfactory surgical outcomes.

## 2. Materials and methods

### 2.1 General information

The study was approved by the Ethics Committee of Guangdong Hospital of Traditional Chinese Medicine (the Second Clinical Medicine College of Guangzhou University of Chinese Medicine, No. 20201127–021), and informed written consent was obtained from all the participants.

This study was conducted at October 2021. We retrospectively reviewed the clinical data of patients with LDH from November 2020 to July 2021 in Zhuhai Branch of Guangdong Traditional Chinese Medicine Hospital. All patients signed an informed consent to have data from their medical records used in research. The authors had no access to information that could identify individual participants during or after data collection.

70 patients with LDH were included in this study, including 48 males and 22 females with an average age of 48 years (16–75 years). All patients underwent PELD were divided into two groups according to the surgical approaches for foraminoplasty. Patients received the conventional foraminoplasty via the multi-step trephine system in the multi-step (MS) group and underwent the one-step foraminoplasty via a large trephine in the one-step (OS) group.

The inclusion criteria are as follows: single-segmental LDH at the L4-L5 level with concordant clinical manifestations, imaging evidence, and physical examination signs; initial surgery; failed conservative treatments of at least 1 month. The exclusion criteria are as follows: multi-segmental LDH; severe lumbar spinal stenosis; severe lumbar spondylolisthesis (degree II and above); scoliosis (Cobb angle > 20˚); infected skin lesions near the surgical site; patients with the poor general condition or major organ disorders who cannot withstand the surgical trauma; elderly patients with multiple underlying diseases and loss of self-care ability; patients who cannot cooperate with the preoperative preparation and postoperative management. Routine preoperative examinations are as follows: lumbar anteroposterior and lateral X-ray,

bilateral oblique x-ray, and flexion-extension x-ray were carried out to measure the height of iliac spines and the vertebral and intervertebral morphology, determining the puncture sites and directions and detecting the lumbar stability; lumbar CT and MRI were conducted to detect the location and degree of disc herniation and detect the calcification.

## 2.2 Surgical procedures

All procedures were performed by the same group of surgeons experienced in both surgical approaches. The patients were placed in the prone position with knees and hips flexed. The lesioned intervertebral space was located using anteroposterior and lateral C-arm fluoroscopy. The puncture point, marked with a marker, was selected at 10–13 cm from the spine midline in a slight cephalic direction and was adjusted according to the patient's physiques. A total of 40 ml anesthetics was prepared with 1% ropivacaine 10 ml, 2% lidocaine 10 ml, and 0.9% saline 20 ml. The puncture was performed with a puncture guide needle at the marked puncture point under local anesthesia. X-ray showed the guide needle was near the midline of the spine in the anteroposterior view and at the posterior horn of the endplate posterior to the vertebral body in the lateral view. The guide needle core was then removed, and a guidewire was placed under local infiltration anesthesia. An approximately 8 mm incision was then made at the puncture point.

Surgical procedures are as follows for patients in multi-step (MS) group (conventional foraminoplasty via the multi-step trephine system): The primary guide rod and the dilating cannula were placed along the guidewire towards the articular process, and a primary trephine (diameter: 5.0 mm) was used to remove the hyperplastic bone at the distal end of the articular process. The position of the trephine was confirmed by fluoroscopy. Then, the primary trephine was removed, the secondary dilating cannula was placed, and a secondary trephine (diameter: 6.5 mm) was used to enlarge the intervertebral foramen. The fluoroscopy was carried out again to confirm the position of the trephine. The secondary trephine was then removed, and the tertiary dilating cannula and a tertiary trephine (diameter: 7.5 mm) were placed to enlarge the intervertebral foramen further. The fluoroscopy was carried out again to confirm the position of the trephine, and the working cannula was placed. The endoscope was later placed and connected to the imaging system. The anatomy was confirmed under the endoscope, and the protruding nucleus pulposus was revealed and removed. Hemostasis and fibrous annuloplasty were performed with a plasma radiofrequency knife. After confirming normal dural sac pulsation and loosed nerve roots, the cannula was removed. The surgery was then completed with suturing of the surgical wounds.

Surgical procedures are as follows for patients in the one-step (OS) group (one-step foraminoplasty via a large trephine): The primary guide rod and a dilating cannula were placed along the guidewire, and a tongue-shaped protective sheath was then placed along the dilating cannula. The opening of the sheath was seen under fluoroscopy to contain the ventral part of the superior articular process (SAP) of the inferior vertebral body. The primary guide rod was kept while the dilating cannula and guidewire were removed. The local infiltration anesthesia was conducted through the tongue-shaped protective sheath at the articular process. A tertiary trephine (diameter: 7.5 mm) was placed and rotated clockwise under the guidance of the primary guide rod, enlarging the intervertebral foramen by cutting off bone parts of the SAP. Once the primary guide rod started to rotate with the trephine, it was indicated that the articular process bone had been removed, and then the position of the trephine was confirmed by fluoroscopy. The fluoroscopy showed that the anterior part of the trephine did not cross the medial edge of the pedicle projection in the anteroposterior view, and the anterior part of the trephine remained at exactly the posterior horn of the endplate in the lateral view. Then the

trephine was removed, and a working cannula was placed. The endoscope was later placed, and the following surgical procedures under the endoscope were the same as described in group MS.

## 2.3 Observation indicators

All the data were collected by another group of physicians who are blind to the patient grouping.

**2.3.1 Perioperative indicators.**  Indicators including the operative time, the time to establish the working cannula, intraoperative fluoroscopy times, the radiation dose, and postoperative complications were recorded. The radiation dose was collected by a radiation dose collector worn on the surgeon's finger.

**2.3.2 Clinical outcome evaluation.**  Visual analogue scale (VAS) was used to evaluate the degree of low back pain and leg pain. Oswestry disability index (ODI) was applied to evaluate the low back function. VAS and ODI were assessed 1 day before the surgery and 2 days and 3 months after the surgery, respectively. LDH in the operated segment with related symptoms on the same side is classified as postoperative recurrence.

## 2.4 Statistical analysis

Statistical analysis was performed with SPSS 26.0, and statistical graphs were drawn using GraphPad Prism 8.0. Quantitative data were presented as mean ± standard deviation. Quantitative data from multiple time points were compared by ANOVA. Rates were compared by the chi-square test. $P < 0.05$ indicated that the difference was statistically significant.

# 3. Results

## 3.1 General condition

All patients underwent PELD surgery successfully. The schematic illustration of foraminoplasty with a single large trephine is shown in Fig 1, and intraoperative fluoroscopy by a C-arm is shown in Fig 2.

Two typical cases are shown in Figs 3 and 4.

All patients were followed up for at least three months. The patient demographic data are shown in Table 1.

## 3.2 Perioperative indicators

All patients underwent PELD surgery successfully. The average operative time in group OS was 100 minutes, significantly shorter than in group MS ($P < 0.01$). The time to establish the working cannula, the intraoperative fluoroscopy times, and the radiation dose in group OS were significantly shorter or smaller than group MS ($P < 0.01$), as shown in Table 2. One case of hyperalgesia and two cases of numbness in the affected limb occurred after surgery in group MS; two cases of burning-like radicular pain and two cases of numbness in the affected limb occurred after surgery in group OS. These symptoms were relieved after using hormones and mannitol, and there was no statistical difference between the two groups in the incidence of complications ($P = 1.000$). No patient underwent secondary surgery during hospitalization in either group.

## 3.3 Clinical outcome evaluation

The postoperative VAS scores and ODI scores (2 days and 3 months after the surgery) were significantly lower than the preoperative scores in both groups ($P < 0.01$). The VAS and ODI

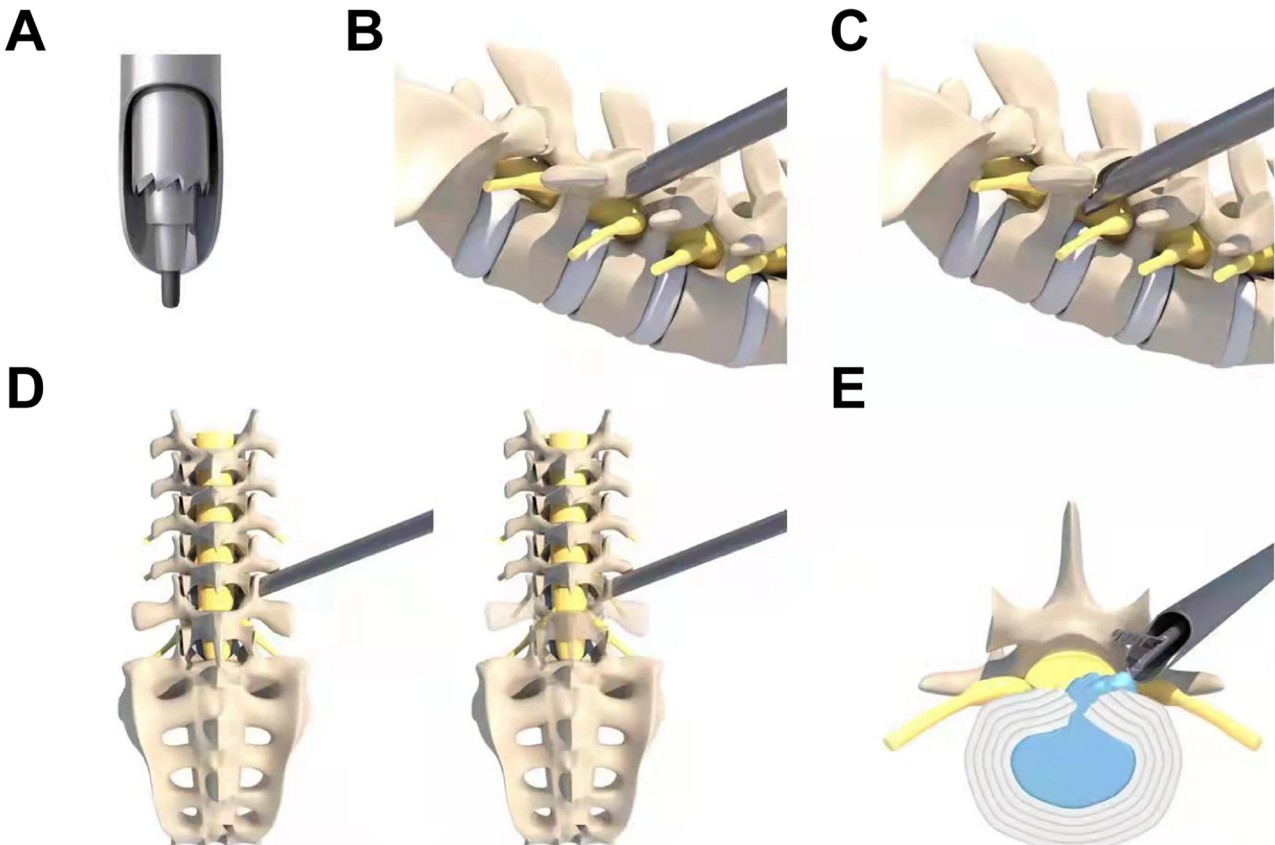

**Fig 1. The schematic illustration of foraminoplasty with a single large trephine.** (A) Position diagram of the guide rod, the dilating cannula, the trephine and the tongue-shaped protective sheath. (B) Lateral position of the dilating cannula. (C) One-step foraminoplasty via a large trephine. (D) Anteroposterior diagram of the dilating cannula and foraminoplasty via a large trephine. (E) Removal of the nucleus pulposus.

scores assessed 3 months after the surgery were significantly lower than those assessed 2 days after the surgery (P < 0.05). There was no significant difference in VAS scores or ODI scores between the two groups at the same time points (P > 0.05) (Fig 5). 3 months after the surgery, recurrence occurred in 3 patients in group MS and 2 patients in group OS. There was no statistical difference in the recurrence rate between the two groups (P = 1.000).

## 4. Discussion

PELD is a typical MISS technique for the treatment of LDH [6], which directly removes the lesioned nucleus pulposus and decompresses the nerve roots under endoscopy, causing no damage to the lumbar back muscles or the posterior ligament complex of the lumbar spine. Meanwhile, it shows little impact on the spine stability and exhibits advantages including less trauma, improved surgical outcomes, and fast recovery [7]. PELD can be completed under local anesthesia, providing surgical opportunities for patients who cannot tolerate the risk of general anesthesia [8, 9]. The classic techniques of PELD include Yeung endoscopy spine system (YESS) and transforaminal endoscopic spine system (TESSYS) techniques [10]. Yeung [11] was the first to access the intervertebral disc through the transforaminal region (Kambin triangle) and gradually remove the disc tissues from within the disc outwards, performing foraminoplasty with the assistance of a high-speed drill. Hoogland et al. [12] addressed the limitations of YESS and designed a set of foraminal reamers with different diameters to remove the

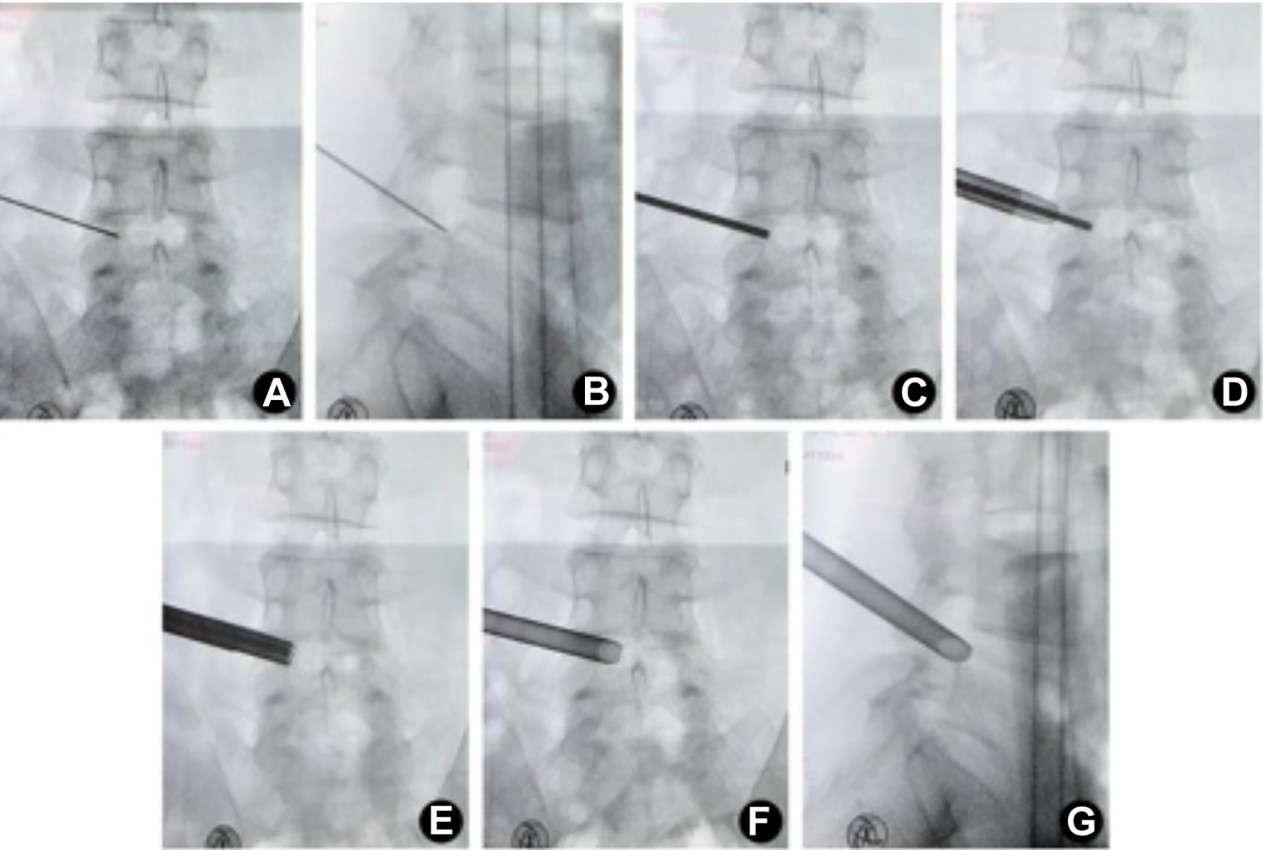

**Fig 2. Intraoperative fluoroscopy.** (A) The anteroposterior fluoroscopy after placing the puncture needle. (B) The lateral fluoroscopy after placing the puncture needle. (C) The fluoroscopy after placing the primary guide rod. (D) The fluoroscopy after placing the tongue-shaped protective sheath. (E) The anteroposterior fluoroscopy after placing the working cannula. (F) The lateral fluoroscopy after placing the working cannula.

bony structure at the inferior anterior margins of SAP step by step and gradually enlarge the foramen. The intervertebral foramen is the necessary access for the placement of percutaneous endoscope [13], but the SAP of the inferior vertebral body is the major bony structure that blocks the establishment of the access route. However, SAP also protects the nerve roots and spinal cord in foraminoplasty [14]. Therefore, the key to completing PELD is enlarging the foramen without damaging the nerves and spinal cord [15]. The TESSYS technique is performed through an enlarged intervertebral foramen to access the spinal canal, which allows easy placement of the surgical working cannula and does not pass through the narrow Kambin's triangle, avoiding damage to outlet nerve roots and ganglia during puncture and cannula placement [16]. The traditional YESS and TESSYS techniques, both of which use multi-grade trephines (three grades) or bone drills (five grades) for foraminoplasty, directly resect bones from SAP in a gradual step-by-step resection way with cumbersome surgical procedures, long learning curves, and the repeated use of multiple trephines, increasing the possibility of nerve injury [17, 18]. In addition, repeated X-ray fluoroscopies are required to confirm the position of trephines in each stage, exposing operators and patients to high radiation [19].

The one-step foraminoplasty via a large trephine is an optimized technique evolving from the conventional foraminoplasty with multi-step trephines, showing significant superiority in the simplified operation. The key to the successful one-step foraminoplasty via a large trephine was the placement of the primary guide rod, which determined the direction and angle of the

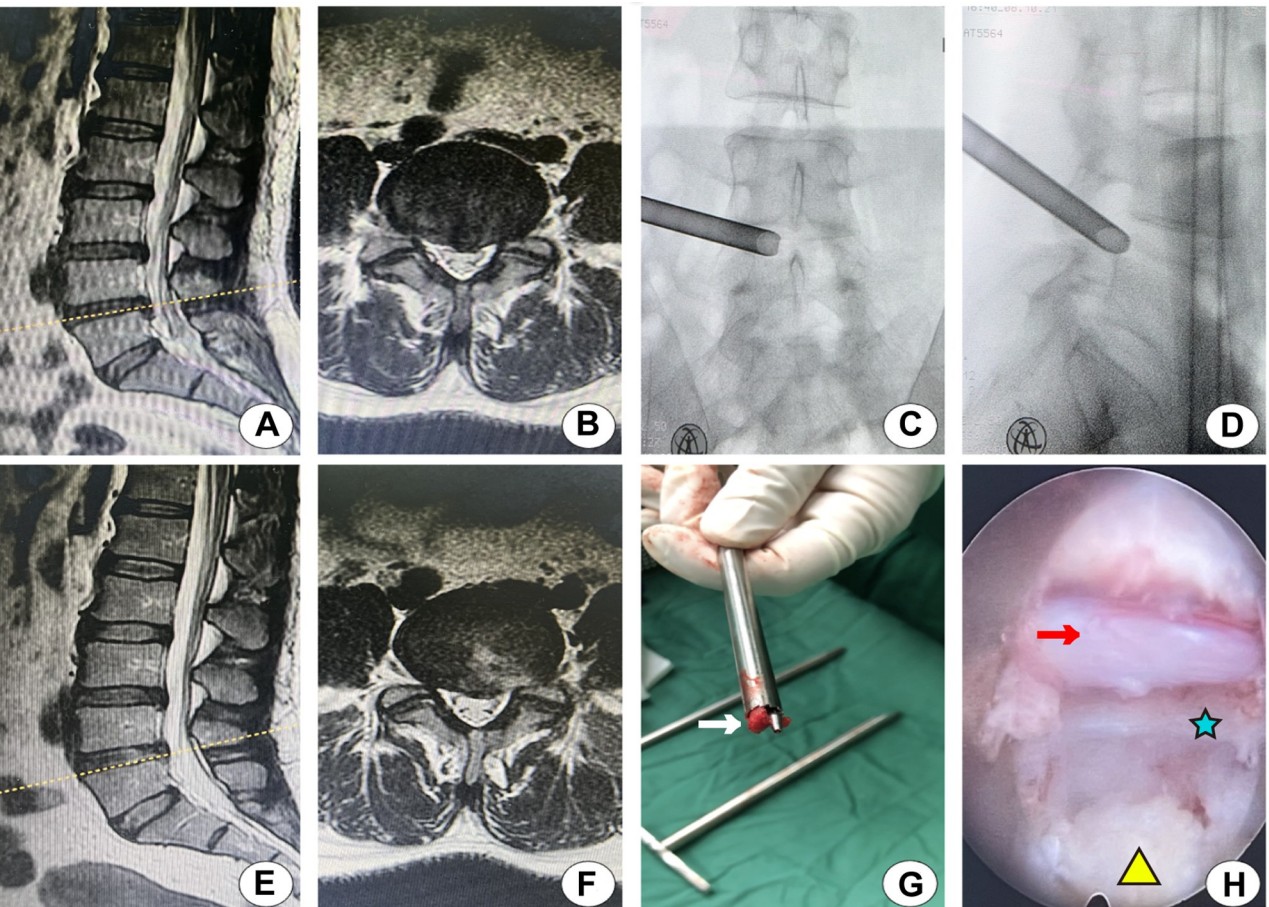

**Fig 3. Typical case one.** (A, B) Preoperative MRI showed the disc prolapse at the L4-L5 level. (C, D) The anteroposterior and lateral fluoroscopy of the working cannula in foraminoplasty with a single large trephine. (E, F) Postoperative MRI showed the complete removal of the protruded nucleus pulposus. (G) The bone parts of the articular process were removed in foraminoplasty. (H) The nerve roots were fully decompressed during the operation: the red arrow, nerve root; the blue star, the posterior longitudinal ligament; the yellow triangle, the nucleus pulposus.

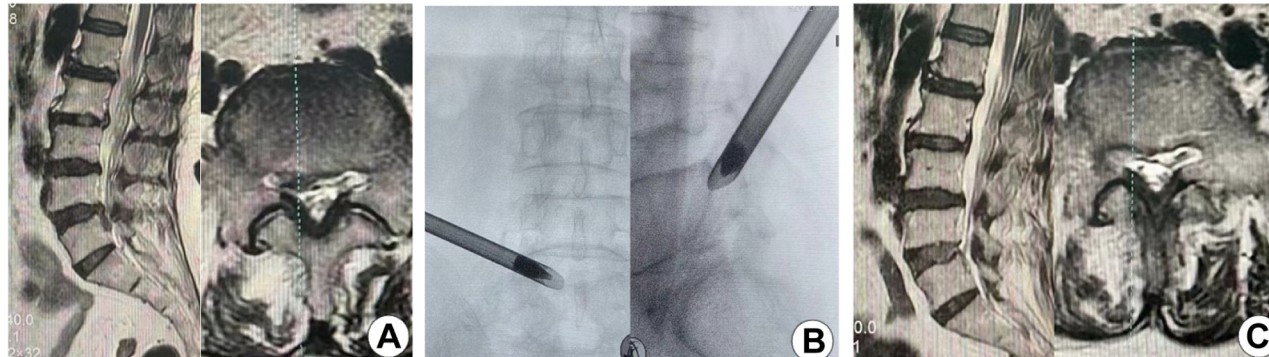

**Fig 4. Typical case two.** (A) Preoperative MRI showed the disc prolapse at the L4-L5 level. (B) Intraoperative anteroposterior and lateral fluoroscopy on the working cannula. (C) Postoperative MRI showed the complete removal of the protruded nucleus pulposus.

**Table 1. The patient demographic data.**

| Group | Gender | | P value | Age | P value |
|---|---|---|---|---|---|
| | Male | Female | | | |
| MS | 22 | 13 | 0.440 | 48.5±15.2 | 0.995 |
| OS | 26 | 9 | | 48.7±14.4 | |

**Table 2. Perioperative indicators.**

| | MS | OS | P value |
|---|---|---|---|
| Operative time (min) | 115.7±25.5 | 100.0±13.1 | <0.01 |
| Time to establish the working cannula (min) | 25.4±6.4 | 15.0±4.0 | <0.001 |
| Intraoperative fluoroscopy times | 20.6±4.2 | 11.5±2.0 | <0.001 |
| Radiation dose ($10^{-5}$Gy/h) | 96.2±19.6 | 53.6±9.5 | <0.001 |
| Complications (n) | 3 | 4 | 1.000 |

trephine into the intervertebral foramen. The followed placement of the dilating cannula dilated the soft tissues and avoided interference with the trephines. As needed, either the conventional multi-grade dilating cannulas or a single dilating cannula can be adopted in the surgery. The use of the single dilating cannula with a diameter of 6.5 mm in the OS group in this study saved time for cannula placement and did not require fluoroscopy. A tongue-shaped protective sheath with a diameter of 8.5 mm was placed to contain the ventral part of the inferior SAP, fixing the trephine access and avoiding the slip of trephines. After removing all the dilating cannulas, the primary guide rod was tapped with an orthopedic hammer to anchor it properly, and a large trephine was then used to cut the articular process. Since there was a space between the trephine and the guide rod, the trephine could directly cut off the bone structure of SAP and embed it in the trephine cannula. The bone part was anchored with the primary guide rod and was easily removed. Therefore, fewer bone fragments were produced and were completely removed with the one-step trephine, effectively reducing the probability of dural and nerve root injury. A larger intervertebral foramen was also obtained, increasing

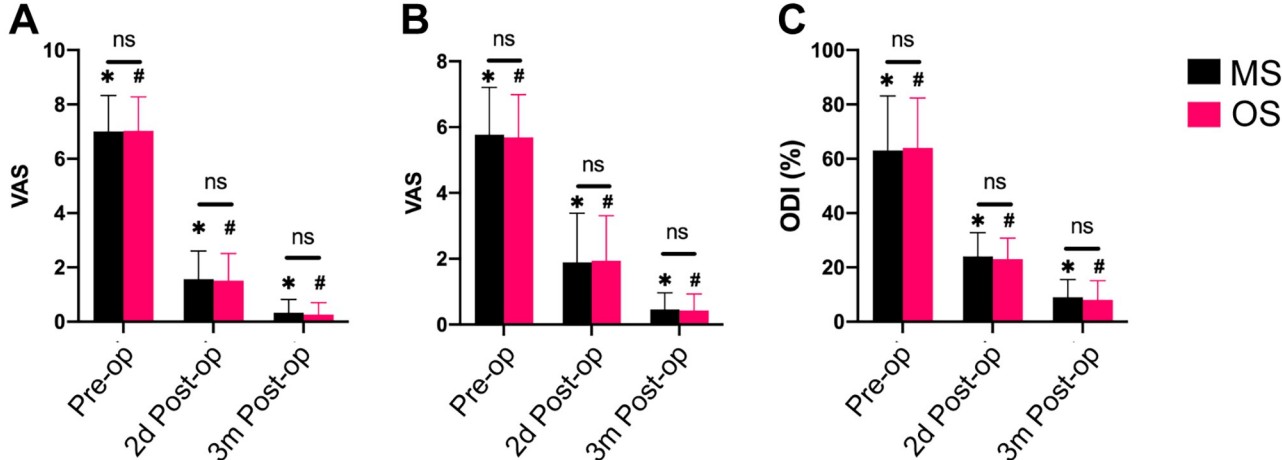

**Fig 5. Clinical outcome.** (A) VAS score of leg pain. (B) VAS score of low back pain. (C) ODI score. ns, no statistical difference; *P < 0.01 (significant difference among different time points of MS group); #P < 0.01 (significant difference among different time points of OS group).

the visual field and surgical space. Meanwhile, the space between the trephine and the guide rod allowed adjusting the direction of the trephine by controlling the position of the tongue-shaped protective sheath. The trephine could be controlled to be closer to the ventral part of SAP to cut off more bone parts and enlarge the cannula space. Notably, the trephine and the working cannula would not slip or shift due to the fixation of the tongue-shaped protective sheath. In the conventional foraminoplasty via the multi-step trephine system, no space between the guide rod and the corresponding trephine was left, making it difficult to adjust the trephine direction during foraminoplasty. Therefore, the conventional foraminoplasty required multiple steps to gradually cut off the articular process and enlarge the intervertebral foramen.

Due to the application of local anesthesia instead of general anesthesia, pain-relieving measures should be considered. Local infiltration anesthesia was performed locally on the articular process through a tongue-shaped protective sheath before the placement of the trephine, which could greatly reduce the pain caused by bone cutting. If obvious pain occurred, the amount of local anesthetic was increased appropriately. In cases of severe nerve root compression by herniated discs, patients usually had severe radicular pain symptoms, which could exacerbate the pain during foraminoplasty because of excessive compression of the discs [20]. Thus, the primary guide rod was adjusted away from the protrusion, and the trephine was operated slowly. It was difficult to adjust the direction during foraminoplasty with the multi-step trephine system. Thus patients' pain could not be relieved, and other anesthetics were required.

The primary guide rod was used as an identifying mark when removing the articular process bone in foraminoplasty via a single large trephine. When operating the trephines, surgeons paid great attention to the movement of the primary guide rod to ensure that the depth border of the trephine was within the connecting line between the inner edges of the upper and lower pedicles. However, for patients with obvious articular process hyperplasia, the width of foraminoplasty might be insufficient, leading to residual ventral bones of SAP and great difficulty in removing the integrated bone block. These patients might need a second operation, thereby prolonging the operation time. Therefore, when the head of the trephine was reaching the connecting line between the inner edges of upper and lower pedicles under anteroposterior fluoroscopy, surgeons slowly reduced the force to operate the trephine and slowly rotated it. Once the primary guide rod rotated with the trephine, which indicated that the bone had been removed, the trephine was rotated several times reversely. After pulling out the trephine with the primary guide rod, the bone parts were seen to embed in the trephine, suggesting the successful bone part removal of SAP, and therefore the working cannula could be established quickly.

For different types of LDH, the angle of the trephine could be guided by adjusting the position of the primary guide rod [21]. In cases when the protruded disc was downward migrated (even when reaching the inner edge of the lower pedicle), the cephalic tilt angle of the primary guide rod was increased to lead the trephine to cut off the bone parts at the upper edge of the pedicle directly. In cases with central disc herniation, the primary guide rod was adjusted to a horizontal position so that the surgical operation area could cover the contralateral nerve roots after trephine placement, thereby avoiding the residual of nucleus pulposus. In cases when the protruded disc was upward migrated (even when reaching the inner edge of the upper pedicle), the primary guide rod was adjusted to a horizontal position with a slight caudal tilt. The caudal tilt angle should not be too large to avoid damage to the outlet nerve roots. Then under the endoscope, the intervertebral disc was cleaned, and the surrounding soft tissue was separated. The angle of the working cannula was gradually adjusted towards the cephalic side to reach the target angle.

There are certainly some limitations in our current research. The present research was a single-center study with a small sample size, which limited its clinical benefits to a certain extent. Large sample multicenter studies are further needed to provide better evidence for future clinical decision-making.

## 5. Conclusion

In this study, a large trephine (the tertiary trephine in the conventional multi-step trephine system) was used for one-step foraminoplasty. The time for foraminoplasty, the total operative time, the intraoperative fluoroscopy times, and the radiation dose in the one-step foraminoplasty were significantly shorter or lower than the conventional multi-step foraminoplasty. The operation procedures have been significantly optimized and simplified, with shorted learning curves and operation time, reduced surgery risk, and reduced X-ray exposure times on surgeons and patients. The foraminoplasty with a single large trephine is worthy of clinical promotion and popularization.

## Supporting information

**S1 Data.**
(XLSX)

**S2 Data.**
(XLSX)

## Author Contributions

**Data curation:** Yao Lu.

**Formal analysis:** Yan An.

**Supervision:** Yong Li.

**Writing – original draft:** Zhaoyu Yu.

**Writing – review & editing:** Bo Wang.

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
