## [Editor Report · Decision Letter 0]

4 Apr 2022

PONE-D-21-40711A one-step foraminoplasty via a large trephine in percutaneous endoscopic transforaminal discectomy for the treatment of lumbar disc herniationPLOS ONE

Dear Dr. wang,

Thank you for submitting your manuscript to PLOS ONE. After careful consideration, we feel that it has merit but does not fully meet PLOS ONE’s publication criteria as it currently stands. Therefore, we invite you to submit a revised version of the manuscript that addresses the points raised during the review process.

We look forward to receiving your revised manuscript.

Kind regards,

L.Miguel Carreira, PhD, MSc,DTO,Ps-Grd,DMD,DVM

Academic Editor

PLOS ONE

Journal Requirements:

a) Did participants provide their written or verbal informed consent to participate in this study?

4. Please ensure that you refer to Figure 5 in your text as, if accepted, production will need this reference to link the reader to the figure.

Additional Editor Comments (if provided):

Dear Authors, 

I hope you are well. I thank you for the opportunity to review this exciting manuscript. I enclose the comments to the manuscript with ID PONE-D-21-40711 , entitled “A one-step foraminoplasty via a large trephine in percutaneous endoscopic transforaminal discectomy for the treatment of lumbar disc herniation ” submitted for publication at  PLOS ONE, after a careful reading. The article raises an important issue related to a more minimally invasive surgical approach for the treatment of LDH.

Perhaps, it would be very interesting if you could offer to readers a side-by-side illustrated comparison between both the treatment procedures. 

Also, please replace the following:

Line 98 - surgical area; should be replaced by surgical site;

Line 333 - with central dis herniation should be replaced by central disc herniation

Considering this, I think the manuscript is ready for publication.

Best regards

---

## [Author Response · Author response to Decision Letter 0]

23 Apr 2022

Reply to the reviewer’s comments

1. Perhaps, it would be very interesting if you could offer to readers a side-by-side illustrated comparison between both the treatment procedures. 

RE: Thanks for the reviewer’s positive comments. We offered an illustration of the new procedures and it would be more interesting to offer the illustration of the traditional procedures side-by-side. However, the traditional step-by-step intervertebral foramen forming technology has been used for many years as a mature technical, which is well known to spinal surgeons. In view of this, considering the costs of time, we are sorry that we didn’t offer the compared illustration.

2. Please replace the following:

Line 98 - surgical area; should be replaced by surgical site;

Line 333 - with central dis herniation should be replaced by central disc herniation

RE: Thanks for the detailed suggestion. We are so sorry for the errors and the word mentioned above has been replaced.

Reply to the Journal Requirements:

RE: Thank you. The revised manuscript has been checked to meet the PLOS ONE’s style requirements.

a) Did participants provide their written or verbal informed consent to participate in this study?

RE: The written informed consent was obtained from the participations and the ethics statement has been amended. 

RE: The full ethics statement has been added in the ‘Methods’ section in the revised manuscript.

4. Please ensure that you refer to Figure 5 in your text as, if accepted, production will need this reference to link the reader to the figure.

RE: Sorry for the mistake. The ‘Figure 2’ in the ‘3.3 Clinical outcome evaluation’ part has been corrected to ‘Figure 5’.

RE: Thank you. The Data Availability statement has been updated and the study’s minimal underlying data set was uploaded as the Supporting Information. 

RE: Thank you. The reference list has been reviewed.

---

## [Editor Report · Decision Letter 1]

3 May 2022

A one-step foraminoplasty via a large trephine in percutaneous endoscopic transforaminal discectomy for the treatment of lumbar disc herniation

PONE-D-21-40711R1

Dear Dr. wang,

We’re pleased to inform you that your manuscript has been judged scientifically suitable for publication and will be formally accepted for publication once it meets all outstanding technical requirements.

Kind regards,

L.Miguel Carreira, PhD, MSc,DTO,Ps-Grd,DMD,DVM

Academic Editor

PLOS ONE
---

## [Editor Report · Acceptance letter]

13 May 2022

PONE-D-21-40711R1 

A one-step foraminoplasty via a large trephine in percutaneous endoscopic transforaminal discectomy for the treatment of lumbar disc herniation 

Dear Dr. Wang:

I'm pleased to inform you that your manuscript has been deemed suitable for publication in PLOS ONE. Congratulations! Your manuscript is now with our production department. 

Kind regards, 

on behalf of

Prof.Dr. L.Miguel Carreira 

Academic Editor

PLOS ONE